# Enhancement of SARS-CoV-2 Infection via Crosslinking of Adjacent Spike Proteins by N-Terminal Domain-Targeting Antibodies

**DOI:** 10.3390/v15122421

**Published:** 2023-12-13

**Authors:** Tina Lusiany, Tohru Terada, Jun-ichi Kishikawa, Mika Hirose, David Virya Chen, Fuminori Sugihara, Hendra Saputra Ismanto, Floris J. van Eerden, Songling Li, Takayuki Kato, Hisashi Arase, Matsuura Yoshiharu, Masato Okada, Daron M. Standley

**Affiliations:** 1Department of Genome Informatics, Research Institute for Microbial Diseases, Osaka University, 3-1 Yamadaoka, Suita 565-0871, Japanhendra@biken.osaka-u.ac.jp (H.S.I.); slli@biken.osaka-u.ac.jp (S.L.); 2Department of Biotechnology, Graduate School of Agricultural and Life Sciences, The University of Tokyo, 1-1-1 Yayoi, Tokyo 113-8657, Japan; tterada@g.ecc.u-tokyo.ac.jp; 3Cryo-EM Structural Biology, Institute for Protein Research, Osaka University, 3-1 Yamadaoka, Suita 565-0871, Japan; kishijun@kit.ac.jp (J.-i.K.); mhirose@protein.osaka-u.ac.jp (M.H.); tkato@protein.osaka-u.ac.jp (T.K.); 4Department of System Immunology, Immunology Frontier Research Center, Osaka University, 3-1 Yamadaoka, Suita 565-0871, Japan; david@biken.osaka-u.ac.jp (D.V.C.); fvaneerden@ifrec.osaka-u.ac.jp (F.J.v.E.); 5Center for Infectious Disease Education and Research, Osaka University, Osaka 565-0871, Japan; matsuura@biken.osaka-u.ac.jp; 6Core Facility, Research Institute for Microbial Diseases, Osaka University, 3-1 Yamadaoka, Osaka 565-0871, Japan; fsugihara@mail.biken.or.jp; 7Department of Immunochemistry, Research Institute for Microbial Diseases, Osaka University, 3-1 Yamadaoka, Suita 565-0871, Japan; arase@biken.osaka-u.ac.jp; 8Department of Immunochemistry, Immunology Frontier Research Center, Osaka University, 3-1 Yamadaoka, Suita 565-0871, Japan; 9Center for Advanced Modalities and DDS, Osaka University, 2-8 Yamadaoka, Suita 565-0871, Japan; okadam@biken.osaka-u.ac.jp

**Keywords:** infection enhancing antibody, cross-linking, SARS-CoV-2, spike protein, molecular dynamics, cryo-EM

## Abstract

The entry of SARS-CoV-2 into host cells is mediated by the interaction between the spike receptor-binding domain (RBD) and host angiotensin-converting enzyme 2 (ACE2). Certain human antibodies, which target the spike N-terminal domain (NTD) at a distant epitope from the host cell binding surface, have been found to augment ACE2 binding and enhance SARS-CoV-2 infection. Notably, these antibodies exert their effect independently of the antibody fragment crystallizable (Fc) region, distinguishing their mode of action from previously described antibody-dependent infection-enhancing (ADE) mechanisms. Building upon previous hypotheses and experimental evidence, we propose that these NTD-targeting infection-enhancing antibodies (NIEAs) achieve their effect through the crosslinking of neighboring spike proteins. In this study, we present refined structural models of NIEA fragment antigen-binding region (Fab)–NTD complexes, supported by molecular dynamics simulations and hydrogen–deuterium exchange mass spectrometry (HDX-MS). Furthermore, we provide direct evidence confirming the crosslinking of spike NTDs by NIEAs. Collectively, our findings advance our understanding of the molecular mechanisms underlying NIEAs and their impact on SARS-CoV-2 infection.

## 1. Introduction

The SARS-CoV-2 virus, which has infected more than half of the human population, enters host cells when the spike protein binds to host angiotensin-converting enzyme 2 (ACE2) [1]. The spike–ACE2 interaction requires a conformational change in the spike receptor binding domain (RBD) from the “down” to the “up” conformation [2]. Consistently, many human antibodies that neutralize SARS-CoV-2 infection block the spike–ACE2 interaction directly by targeting the RBD [3,4]. As a consequence, most of the SARS-CoV-2 variants that have emerged can be characterized by their RBD mutations, presumably because such mutations allow the virus to temporarily escape neutralization [5,6]; however, since the spike protein presents a large and complex antigenic surface, a significant proportion of anti-spike antibodies target regions other than the RBD. The functional roles of most non-RBD targeting antibodies are poorly understood.

Among non-RBD targeting antibodies, those that recognize epitopes on the N-terminal domain (NTD) are of particular interest, as three NTDs and three RBDs wrap around each other in order to make up the spike S1 subunit, implying that the NTD and RBD functions are also intertwined. One group of anti-NTD antibodies targeting an epitope that is structurally close to the RBD are themselves neutralizing [7]. The molecular mechanism of this neutralization has not been elucidated; however, the NTDs in other coronaviruses have been shown to interact with host cell molecules such as sialic acid [8] and the SARS-CoV-2 S1 subunit interacts with heparan sulfate [9] in a manner that enhances infection [10]. Thus, the fact that antibodies binding to the surface of the NTD proximal to the host cell can block infection seems reasonable.

Yet, another group of anti-NTD antibodies has been shown to enhance infection [11,12]. Interestingly, these NTD-targeting infection-enhancing antibodies (NIEAs) target an overlapping surface patch that is distal to the host cell. Antibody-dependent infection enhancement (ADE) has been well-studied in several other viruses, including Dengue, HIV, Measles virus, and RSV [13,14,15,16,17]. Several studies also showed in vitro ADE in SARS CoV-2, SARS CoV, and MERS CoV infection [18,19,20]. However, all previously reported ADE mechanisms are dependent on the antibody’s fragment crystallizable (Fc) region [13,15,21,22]. In contrast, it was shown that the NIEAs are not dependent on the Fc, implying a novel ADE mechanism [11,12]. ADE can have profound consequences on the safety profile of vaccines, as seen in Sanofi Pasteur’s Dengvaxia, which sensitizes dengue seronegative individuals, resulting in higher risk of severe dengue [23,24]. In addition, because very large proportions of the human population are regularly vaccinated for SARS-CoV-2, it is important to understand the mechanism of this novel form of ADE. In this regard, it is noteworthy that a recent survey conducted by our laboratory showed that the frequency of NIEAs was comparable in COVID-19 patients and COVID-19-vaccinated healthy donors but not healthy unvaccinated donors [25]. Taken together, the ADE mechanism in SARS CoV-2 needs to be studied and evaluated further to be taken into consideration for possible ADE risk in developing SARS CoV-2 vaccines and therapies [26,27].

In this report, we first review the evidence for NIEA-mediated ADE and describe a spike–NIEA–spike crosslinking model that is consistent with this evidence. Next, we present new results from molecular dynamics simulations (MD) and hydrogen–deuterium exchange mass spectrometry (HDX-MS) that more precisely resolve NIEA–NTD interactions at the residue level. We then provide direct observation of NIEA-crosslinked NTDs using negative-stain electron microscopy that is consistent with our structural model. Together, these results provide strong support for the spike–NIEA–spike crosslinking model, which is qualitatively distinct from previously described ADE mechanisms. A recent study reports that 5.9% of anti-spike antibodies in COVID-19 convalescent patients were infection-enhancing [28]. Given the ubiquity of spike proteins in human coronaviruses, along with their prevalence in many other enveloped viruses, it is of interest to understand the properties of NIEAs not only in the context of SARS-CoV-2 but in host–virus interactions in general.

## 2. Material and Methods

Cell lines. Expi293 cells (Thermo Fisher Scientific) were cultured with the HE400AZ medium (Gmep). The cells were routinely checked for mycoplasma contamination.

*Plasmids.* The SARS CoV-2 spike and 2490 monoclonal antibody plasmids were prepared as described previously [5]. The SARS CoV-2 spike NTD (amino acids 14-333) was cloned into a pcDNA3.4 expression plasmid containing a SLAM signal sequence and His-tag at C-terminus. The DNA sequence was confirmed using DNA sequencing (ABI3130xl).

Protein expression and purification. The pcDNA3.4 plasmid containing His-tagged NTD of SARS CoV-2 spike protein was transfected to Expi293T cells using PEI max (Polysciences). After 18–21 h post-transfection, Gxpress 293 Enhancer (Gmep) was added into culture media to enhance translation. Culture supernatant containing His-tagged NTD protein was harvested 4 days after transfection and was purified using Ni-NTA resin (Qiagen). NTD protein was further purified with a Superdex 200 Increase 10/300 GL gel filtration column using AKTA Pure 25 System (Cytiva). The pCAGGS vector containing the 2490 antibody was transfected using PEI into Expi293T cells. Gexpress 293 (Gmep) enhancer was added into culture media after 18–21 h post-transfection. Culture supernatant containing the 2490 antibody was harvested 4 days post-transfection and purified with HiTrap Protein A column using AKTA Pure 25 System (Cytiva) followed by buffer exchange into PBS.

Sample preparation. The NTD protein was incubated with the 2490 antibody at a mass ratio of 5:1 and incubated at room temperature for 16–18 h. The complex was purified on a Superdex 75 Increase 10/300 GL (Cytiva) column equilibrated with 20 mM Tris-HCl pH 8.0, 300 mM NaCl.

Negative staining. An aliquot of 5 µL of purified complex was applied to a glow-discharged, 600-mesh, carbon-coated grid for 30 s. The grid was then blotted with filter paper to remove excess sample. Subsequently, the grid was negatively stained with three drops of 2% uranyl acetate, blotted again with filter paper, and air-dried. Finally, the grid was transferred to a Talos Arctica transmission electron microscope (Thermo Fisher Scientific) equipped with a K2 summit camera and operated at 200 kV. Data were collected in linear mode at a nominal magnification of 28,000, corresponding to a resolution of 1.482 Å/pixel.

All image processing was carried out using cryoSPARC software v4.3.0 [29]. After CTF estimation, about 600 particle images were manually picked to create the image templates. The manually picked images were then subjected to 2D classification into 20 classes. Subsequently, the 2D classes in which the NTD–NIEA–NTD trimer was clearly visible were selected as templates. The particle images picked using the template were 2D, classified into 100 classes, and the particles included in the selected classes were retained. This process was repeated two more times.

Hydrogen–deuterium exchange mass spectrometry (HDX-MS). 2490 antibody was added to NTD solution at 1:2 ratio to make antigen–antibody complex. Both NTD protein and NTD-2490 antibody complex were dispensed in a vial in Tris-HCl buffer (20 mM Tris-HCl, pH 8.0, 300 mM NaCl). HDX was performed on a LEAP HDx-3 PAL platform robot operated with the Chronos software v. 3 (Trajan Scientific and Medical). A total of 2 μL of free NTD protein or NTD-2490 antibody complex was incubated and diluted into the 28 μL of Tris-HCl buffer consisting of 100% H_2_O or 90% D_2_O with an indicated incubation timing. In total, 30 μL of the deuterated samples was mixed with 30 μL of quench buffer 2 M Guanidine hydrochloride, 100 mM citrate in 0.1% formic acid water, to stop the HDX reaction. Then, 50 μL of the sample-quench buffer mixture was injected into the online digestion column, Immobilized Protease XIII + Pepsin column, with 0.1% formic acid water flow rate at 2 µL/min for 6 min.

Digested products were trapped using GL Science Intersil Sulfa C18 guard column 1.0 × 10 mm for 1 min. The product was later eluted and separated by Thermo Scientific Hypersil GOLD C18 2.1 × 50 mm column. Analytical column flow was controlled by Agilent 1290 HPLC pump with mobile phase A: 0.1% formic acid water and B: 0.1% formic acid acetonitrile. The analytical elution gradient was set as 5%B to 40%B in 20 min.

MS analysis was performed using the following conditions: ESI positive, capillary voltage at 4500 V, dry gas at 4.5 L/min, scan range from 350 to 2000 m/z using timsTOF Pro (Bruker). MS2 was conducted for peptide identification, and MS measurements were conducted for non-deuterium and deuterium labeling experiments. The peptide identification was processed using PEAKS Studio X (Bioinformatics Solutions Inc.). Hydrogen–Deuteurium exchange data of NTD protein and NTD-2490 antibody complex were analyzed with HDXaminer v. 3.2.1 (Sierra Analytics).

Antibody modeling. All-atom models of two cross-linked NIEAs (2490 and 8D2) were built using Repertoire Builder, a template-based antibody modeling tool [30]. 

Molecular dynamics (MD) simulations. Simulations were performed for the NIEA Fab–NTD complexes and for the NTD alone. Based on the SARS-CoV-2 full-length prefusion spike trimer structure (PDB ID: 7JJI) [31], 1000 NIEA Fab–NTD complexes were generated using PyRosetta [32], which were subsequently clustered. In total, nineteen representative model structures were selected for six antibodies, and an MD simulation was performed for each model. In the simulations, the N- and the C-termini of the NTD were capped with an acetyl and an *N*-methyl group, respectively. The C-termini of the heavy and the light chains of the NIEA Fabs were capped with an *N*-methyl group. Histidine residues were protonated at the Nε2 atoms, except for His107 of the heavy chain of the NIEA Fab of model 2660_10_1, which were protonated at the Nδ1 atom. Each structural model was immersed in a cubic box of water, ensuring a minimum distance of 10 Å between any box face and any protein atom. Chloride ions were added to neutralize the system. The edge lengths of the cubic box were about 110 Å for the NIEA Fab–NTD complexes and about 95 Å for the NTD alone. The Amber ff14SB force field parameters [33] were used for the proteins and ions and the TIP3P model [34] was used for water. After energy minimization and equilibration, production MD runs were performed for 1 μs. During the MD simulations, the temperature was maintained at 300 K using the velocity-rescaling method [35] and the pressure was maintained at 1.0 × 10^5^ Pa using the Berendsen weak coupling method [36]. Bond lengths involving hydrogen atoms were constrained using the LINCS algorithm [37,38] to allow a time step of 2 fs. Electrostatic interactions were calculated using the particle mesh Ewald method [39,40]. All MD simulations were performed using Gromacs 2020 [41], with coordinates recorded every 10 ps. RMSDs were calculated for antibody Cα atoms and NTD Cα atoms for each snapshot structure after fitting the antibody Cα atoms on those of the last snapshot structure. The RMSD values were averaged over the last 500 ns and the last 300 ns. The average structures were calculated using the snapshot structures of the last 500 ns of the simulations for model 2490_7_1 and all the snapshot structures of the simulation for the NTD alone. RMSFs around the average structures were calculated for Cα atoms. Visualization and rendering of simulation snapshots were performed with the molecular graphics viewers PyMOL [42] and UCSF Chimera [43].

Biolayer Interferometry analysis. Biolayer interferometry was performed using an Octet Red96 instrument (ForteBio, Inc.). SARS-CoV-2 spike protein-biotin conjugated with the concentration of 1 µg/mL was immobilized on a Streptavidin-coated biosensor surface. The baseline was obtained using measurements taken for 60 s in 1× Kinetics buffer (Fortebio); next, the sensors were subjected to association phase immersion for 300 s in wells containing 2490-Fab or 2490-IgG1 diluted in 1× Kinetics buffer. The sensors were immersed in 1× Kinetics buffer for 600 s to measure dissociation. The apparent KD values of the 2490-Fab or 2490-IgG1 binding affinity for spike protein were calculated from all the binding curves based on their global fit to a 1:1 Langmuir binding model.

## 3. Results

### 3.1. Evidence for Spike–Spike Crosslinking by NIEAs

Previously, six different antibodies of independent genetic origin that had been described in the COVID-19 literature were found to enhance ACE2 binding in vitro [11]. The hypothesis that NIEAs exert their enhancing effect by a common mechanism involving cross-linking adjacent spike proteins was originally based on three crucial pieces of evidence.

NIEAs of distinct genetic origin were shown to compete with each other. This observation was supported by mutagenesis, which identified a narrow patch of residues on the NTD distal from the spike–ACE interface. The location of the epitope was further demonstrated by the Cryo-EM studies using fragment antigen-binding regions (Fabs), which revealed a binding mode in which the Fab makes an angle of approximately 30 degrees with the spike axis. The cryo-EM result agreed very well with docking models, as well as with an earlier anti-NTD monoclonal antibody cryo-EM structure, although ACE2 binding enhancement was not reported for the antibody in this structure [11]. Further support came from an independent report of a set of NIEAs whose structural and functional properties matched those described by Liu and co-workers [12]. Due to the dynamic nature of the spike itself, the side chains of interface residues were not resolved in the two published studies of spike-bound NIEAs, leaving open the question of the detailed atomic structure of this interaction. The effect of alanine substitutions on the NIEA-enhancing effect is shown in Figure 1A;The Fc region is dispensable for its enhancing effect. It was demonstrated that while both monovalent Fabs and divalent IgG antibodies, as well as (Fab)2 constructs, exhibited strong binding, only the divalent constructs led to an increase in ACE2 binding. This suggested that the presence of both Fabs is essential for enhancement, with the Fc region playing no significant role (Figure 1B). We further confirmed that the Fc region does not affect binding affinity to the spike using Bio-Layer Interferometry (Figure 1C);Two epitopes on one spike trimer are not accessible by one NIEA. The question of how one NIEA interacts with two epitopes was investigated computationally. Using full-length IgG Protein Data Bank entries or individual-particle electron tomography IgG reconstructions [44] as structural templates, we assessed the feasibility of two Fabs docking to one spike; however, no IgG structures that had both arms docked to two NTD epitopes were observed (Figure 1D). This does not rule out such a conformation, as IgG antibodies might have rarely observed structures that allow intra-spike binding. However, we have been unable to generate a model where one divalent NIEA binds with both Fab arms to one spike.

Taken together, these three observations suggest the possibility that two antibody arms bind NTD epitopes on adjacent spikes.

### 3.2. Structural Model of Spike–NIEA–Spike Crosslinking

In order to investigate the plausibility of a spike–NIEA–spike crosslinking model, we again utilized known IgG structures and the low-resolution NIEA Fab–spike structure. In brief, pairs of Fab–spike complexes were positioned relative to each other by uniformly sampling spherical coordinates specified by the envelope radius and two rotational degrees of freedom. Known IgG antibodies were then fitted onto the two Fabs. Under a given assumed envelope radius, different combinations of the two rotational degrees were sampled. Cross-linked models were then ranked according to the quality of the fit of the two Fabs. Finally, top models with different spike orientations were selected as candidates (Figure 2A). The cross-linked spikes were further modeled in a realistic plasma membrane (Figure 2B). Encouragingly, the spike–spike distance in cross-linked models (23 nm) agreed well with the mean value observed in Cryo-EM tomography of 100 spike pairs in whole virions: 23.6 ± 8.1 nm [45].

There are several known limitations to this model, in spite of its self-consistency. Most importantly, the binding of a divalent IgG antibody to two spikes has not been observed directly. When full-length antibodies were mixed with soluble spikes at a 1:1 ratio, the sample aggregated (data not shown). A second limitation in the above models is that the resolution of the Fab–NTD interface was insufficient to resolve side chains, which, in turn, hinders a detailed biophysical explanation for the enhancing effect. 

### 3.3. Structural Dynamics of the NIEA Fab–NTD Interface

To better resolve the interface of the antibody–NTD complex at atomic resolution, we performed MD simulations using a Fab instead of a full antibody. A total of nineteen 1 µs simulations were performed starting from different docked conformations of the six NIEA Fabs to the NTD. As a reference, a single NTD was also simulated. During the simulations, the Fab and NTD fragments rearranged themselves, optimizing their relative orientation with respect to the docked conformation. The RMSD analysis indicates that some complex models converged to stable conformations with the average RMSD values from the last snapshot structure less than 3 Å (Figure 3A). Figure 3B shows the structural ensembles obtained from the simulation of the NTD alone and from that of model 2490_7_1, which showed the smallest average RMSD value. RMSF calculations showed that both the Fab and the NTD fragments were internally stable (RMSF < 2 Å) in most regions; however, the flexible loops of the NTD (loops 1–3) exhibited large fluctuations. Comparison of the NTD and NTD–Fab simulations showed that antibody binding resulted in a stabilization of NTD loop 1, which contains several residues identified by Liu et al. [5] as belonging to the overlapping epitope.

### 3.4. Change in Solvent Accessibility of the NTD Loops Consistent with MD Modeling

Due to the dynamic nature of the NIEA Fab–NTD complexes, it is difficult to establish the epitopes with high confidence using MD. In order to independently assess the binding interface residues of the published NIEA Fab (2490) and the NTD, we next performed HDX-MS on the NTD alone and NTD + 2490 Fab. A comparison of the hydrogen–deuterium exchange rates was performed. Consistent with the MD simulations, loop 1 (containing residues “SGT”), also referred to as loop β3–β4 [46], exhibited significant differences upon Ab binding (Figure 3C). It is worth pointing out that this region had poor peptide coverage, which was not improved by adjusting the HDX-MS reagents and conditions. However, the protection of the loop 1 region was consistent in multiple observations. Taken together, we concluded that loop 1 is a critical epitope for NIEAs. In this regard, it is noteworthy that the composition of loop 1 in SARS-CoV-2 is completely distinct from that of SARS-CoV and other coronaviruses in subgenus *Sarbecovirus* [47].

### 3.5. NIEAs Crosslink SARS-CoV-2 Spike N-Terminal Domains

In order to validate the crosslinking model, we next examined the full-length IgG antibodies using electron microscopy (EM). As mentioned above, when IgG NIEAs were mixed with full-length spikes at a 1:1 concentration, the proteins aggregated, presumably due to extended crosslinking made possible by the three enhancing epitopes on a given spike trimer. While this interpretation is consistent with the crosslinking model, it does not provide direct independent evidence for the phenomenon. Therefore, we reasoned that observing IgG antibodies in the presence of monomeric spike NTDs would provide such evidence. According to the model, an IgG with a soluble monomeric NTD would be expected to form only NTD–NIEA–NTD trimers.

Consistent with this prediction, the NTD–NIEA–NTD trimers were clearly visible through negative stain imaging. A total of 165,539 particles were picked by template matching. After three rounds of 2D classification, typical 2D class images are shown in Figure 4A. In each class image, the density corresponding to the NTD and Fabs was clearly visible, while that of the Fc was weak and blurry, suggesting that the Fc region is highly flexible. NTDs bound to both Fab arms in all classes. In addition, the angles between the two Fabs varied considerably, indicating the flexibility of the IgG hinge region. Two well-resolved 2D classes are shown, which agree qualitatively with the cross-linked structural model constructed previously (Figure 4B).

## 4. Discussion

ADE mechanisms in viral infections are well-documented, and their mode of action has been attributed to the Fc region. ADE has been observed in several viral infections and is associated with the antibody binding of immune complexes to cellular receptors, including Fc receptors (FcRs). In contrast, the NIEA mode of action is Fc-independent. Using this and other experimental evidence, we concluded that a model that explains the known data is one in which neighboring spikes are cross-linked by NIEAs. Such crosslinking may affect the RBD transition from the “down” to the “up” state by decoupling the NTD–RBD interactions. If true, NIEAs constitute a unique form of ADE.

Previous work resolved the structure of NIEA Fab–spike interactions at low resolution. Here, we extended these findings by first constructing an all-atom model that was shown by MD simulations to stabilize NTD loop 1. Further analysis using HDX-MS confirmed this interaction, suggesting that loop 1 is a crucial conformational epitope for NIEAs. The stabilization of loop 1 raises the possibility that NIEAs may induce conformational changes in the NTDs allosterically. This idea was also suggested by the HDX-MS data, which indicates changes in solvent accessibility far from the epitope upon Fab binding. However, further work is needed to investigate possible allosteric effects.

This study provides direct evidence for the crosslinking of NTDs by NIEAs, as all observed IgGs appeared to be bound to two NTDs in negative-stained images. Moreover, their overall orientation was consistent with that shown in the original cross-linked spike model. This supports the proposed model of spike–spike crosslinking by NIEAs, where neighboring spike proteins are connected by IgGs. As previous research points out that antibodies generated from vaccination or from therapeutic antibodies both have the potential to cause ADE of SARS CoV-2 infection in vitro [18], it is important to check for ADE potential in COVID-19 therapeutics and vaccines. The main open question is how NTD binding by IgG facilitates ACE2 binding. Several studies highlight the interconnectedness and coupling of NTD and RBD [43,48,49]. Moreover, NTD movement is involved in RBD conformational change from the “down” to the “up” position [49]. We have proposed that NTD crosslinking decouples NTD–RBD interactions, potentially facilitating the RBD “down” to “up” transition. This study opens the door to future investigations into the impact of NIEAs on virus behavior, with potential implications for therapeutic and vaccine development.

## Figures and Tables

**Figure 1 viruses-15-02421-f001:**
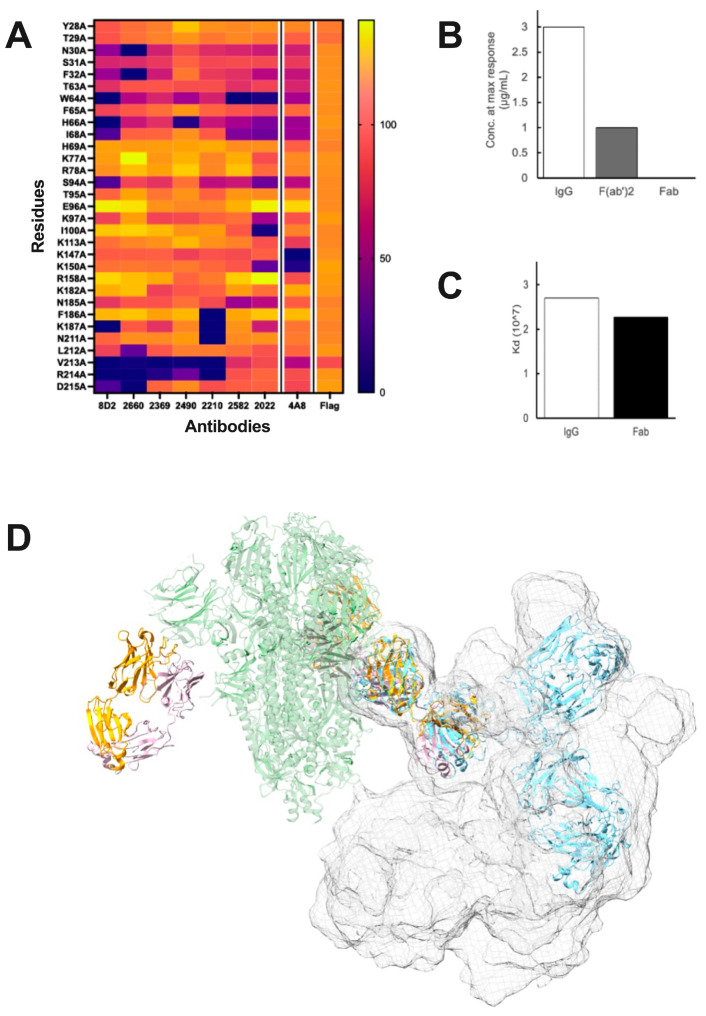
Evidence for Spike-NIEA crosslinking. (**A**), The effect of Alanine substitution on Spike-ACE2 binding, as quantified by mean fluorescence intensities, revealed an overlapping set of epitope residues including W64, H66, K187, R214, and V312. (**B**), Concentration of NIEA 2490 at the maximum ACE2 binding enhancement (N.R. = no response). (**C**), Dissociation constant (Kd) of NIEA 2490 to Spike trimer. Binding affinity assays indicated that the Fc region does not affect ACE2-Spike binding, but that both Fab arms are required. (**D**), IgG structures superimposed on the spike-Fab cryo-EM model indicated that it is unlikely for the second arm of the IgG to reach the NTD epitopes on the other chains in the same spike trimer. Spike (green) complexed with DH 1052 Fab (yellow and pink) (PDB ID: 7LAB), representative IgG pose (blue), cumulative surface representation of IgG poses (surface mesh).

**Figure 2 viruses-15-02421-f002:**
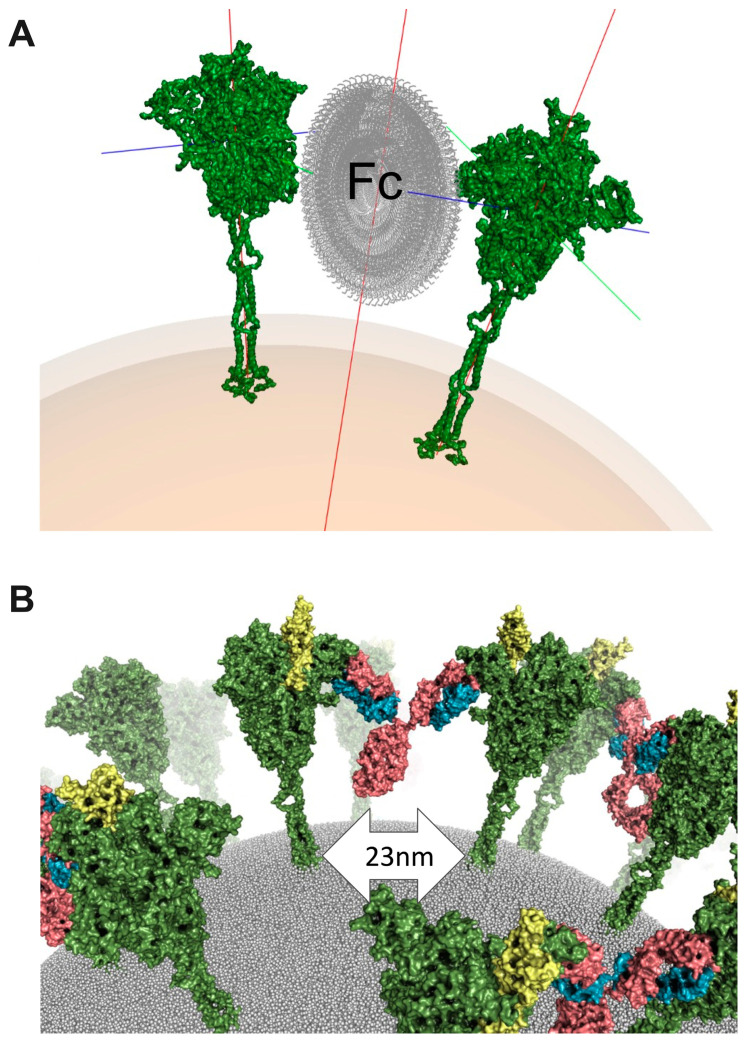
Cross-linked structural model. (**A**), Sampling Fc orientations to guide template-based modeling for spike crosslinking. Spike is colored green, and Fc is colored grey. Spike protein x, y, and z axes are represented by green, red, and blue lines coming from spike protein. Particle center axis is represented by red line. (**B**), A realistic model of crosslinked spike proteins on the virus surface. Spike is colored green and RBD in open conformation is colored yellow. Antibody is colored in pink (heavy chain) and blue (light chain).

**Figure 3 viruses-15-02421-f003:**
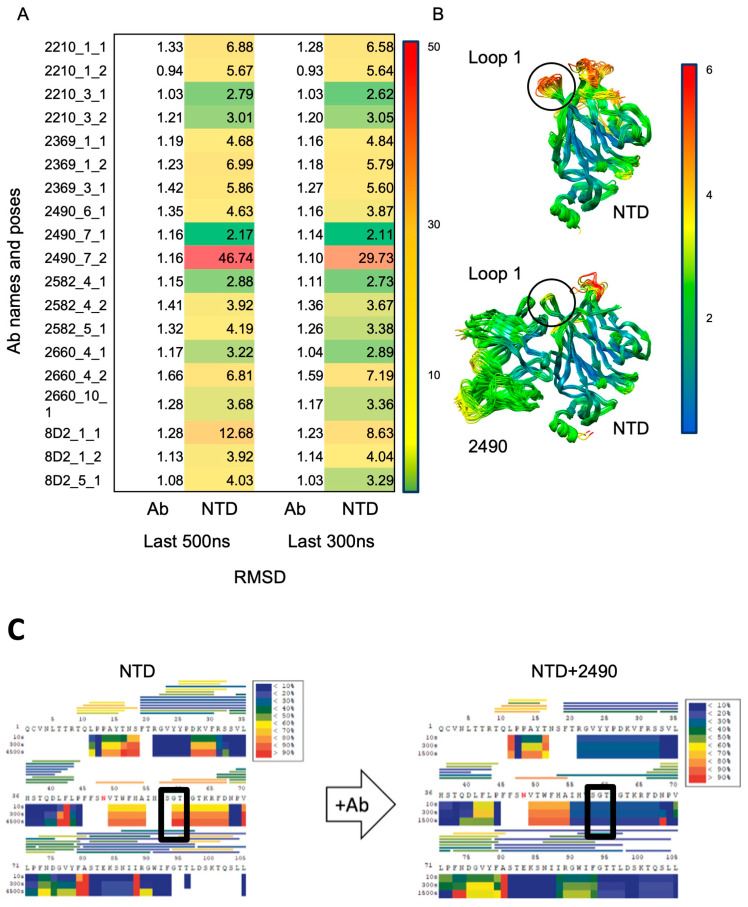
Refinement of NIEA-NTD interface. (**A**), Root-mean-square fluctuations (RMSFs) from initial structure during molecular dynamics (MD) simulations of six NIEAs docked to NTD, in triplicate. (**B**), Table of per-residue RMSF showing that NIEA-bound NTD loop 1 is stabilized. Lower RMSF indicates stability. RMSF score ranging from 0 (green) to 50 (red). (**C**), HDX-MS study of NTD and NTD with NIEA 2490, where the rectangle indicates the change in solvent accessibility around NTD Loop 1 shifting from high to low upon 2490 binding.

**Figure 4 viruses-15-02421-f004:**
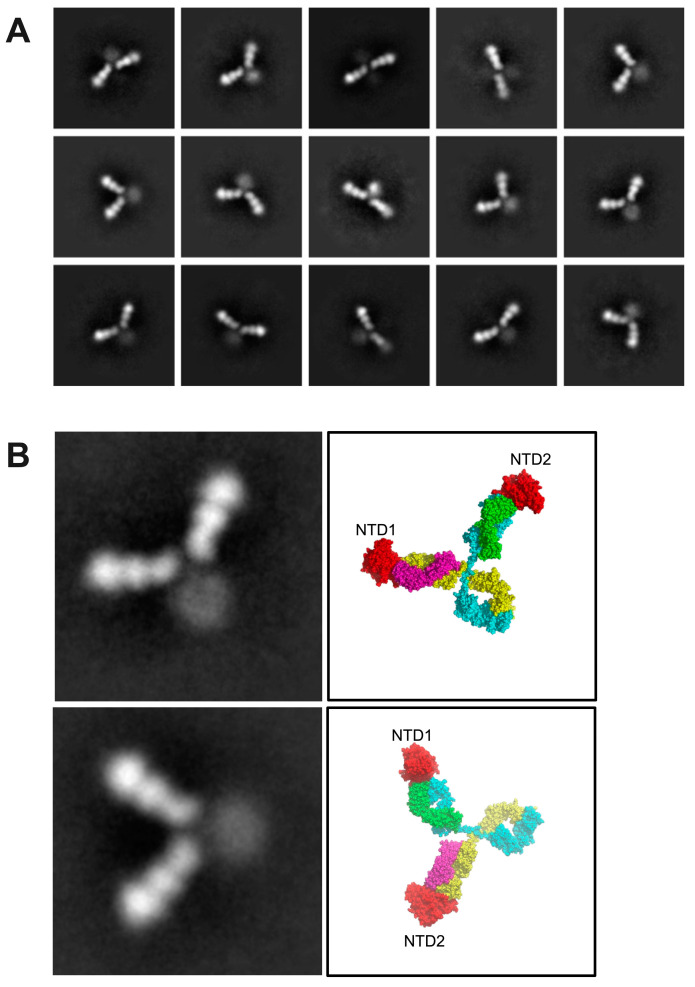
NIEA 2490 cross-links spike NTDs. (**A**), negative stained EM images show a variety of IgG conformations, all of which appear to have two bound NTDs. (**B**), A close-up view of two well-populated 2D classes alongside molecular models indicating the locations of the two bound NTDs. NTDs are colored red, antibody heavy chain is colored blue and yellow while antibody light chain is colored by magenta and green.

## Data Availability

The data presented in this study are available on request from the corresponding author.

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
