# Peer review of "Enhancement of SARS-CoV-2 Infection via Crosslinking of Adjacent Spike Proteins by N-Terminal Domain-Targeting Antibodies"

_viruses, 2023, doi:10.3390/v15122421_

Round 1

Reviewer 1 Report

Comments and Suggestions for Authors

Well written paper.  I have little to say about it, namely:

Figure 3A: what is the meaning of the color code ?

Typos:

l177: of the -> for the

l303: Fig.3B is not a heatmap (not a map).

Comments on the Quality of English Language

Good.

Author Response

Response to Reviewer 1 Comments

  1. Summary

Thank you for giving us the opportunity to submit a revised version of the manuscript “Enhancement of SARS-CoV-2 Infection via Crosslinking of Adjacent Spike Proteins by NTD-Targeting Antibodies” for publication in Viruses. We thank you for taking the time to review this manuscript. We have incorporated suggestions made by reviewer and highlighted those changes within the revised manuscript. Below, please find the detailed responses below and the corresponding revisions highlighted in revised manuscript. All page and line numbers refer to the revised manuscript.

Comments 1: Figure 3A: what is the meaning of the color code?

Response 1: As pointed out by the reviewer, we have modified Figure 3A by adding continuous color legend and annotate the meaning of the color in figure caption.

Figure Caption (Line 366):

RMSF score ranging from 0 (green) to 60 (red).

Comments 2: l177: of the -> for the

Response 2: Thank you for pointing this out. We changed “of the” to “for the” in line 164.

Method (Line 164-166):

Molecular dynamics (MD) simulations. Simulations were performed for the NIEA Fab-NTD complexes and for the NTD alone. Based on the SARS-CoV-2 full-length prefusion spike trimer structure (PDB ID: 7JJI)

Comments 3: l303: Fig.3B is not a heatmap (not a map).

Response 3: Thank you for the correction. We changed the figure caption from “heatmap” to “graphical representation” in line 363.

Figure Caption (Line 366-367):

B, Graphical representation of per-residue RMSF showing that NIEA-bound NTD loop 1 is stabilized.

Reviewer 2 Report

Comments and Suggestions for Authors

This study investigated a novel antibody-dependent enhancement (ADE) mechanism in SARS-CoV-2, distinct from traditional Fc region-dependent processes. It introduced structural models based on molecular dynamics (MD) simulations and hydrogen-deuterium exchange mass spectrometry (HDX-MS), providing comprehensive molecular-level insight into the interactions between NTD-targeting infection-enhancing antibodies (NIEA) and the N-terminal domain (NTD) of the SARS-CoV-2 spike protein. Additionally, the research included direct observations of NIEA-crosslinked NTDs using negative staining electron microscopy, which validated the structural model. Furthermore, the study explored the structural dynamics at the NIEA Fab-NTD interface, revealing the stabilization of specific NTD loops upon antibody binding. This suggested that NIEAs may induce conformational changes in the NTDs, potentially influencing the transition of the receptor-binding domain (RBD) from the “down” to the “up” state.

The study could be enhanced by addressing a few key concerns.

1.     What impact does this research have on the development of vaccines and treatment of COVID-19? The paper would benefit from a more in-depth discussion of the implications of its findings for vaccine and therapeutic antibody development.

2.     The study points out that the composition of NTD loop 1 in SARS-CoV-2 is distinct from that of SARS-CoV-1. Did the authors examine other coronaviruses to determine if this characteristic is exclusive to SARS-CoV-2?

3.     The study suggests that NIEAs may induce allosteric changes in the NTDs, affecting the RBD transition from the “down” to the “up” state. This remains a hypothesis. Could the author offer additional evidence to support the role of these interactions?

Comments on the Quality of English Language

This manuscript was well-written, but there are still some language errors. The authors should carefully review the entire manuscript and make improvements where necessary.

For example, in line 97, "These images clearly show the crosslinking if NTDs in a manner that is consistent with our structural model." The word "if" seems to be a grammar error.

The authors should ensure consistent use of terms throughout the manuscript . For example, in line 79, "NTD-targeting infection enhancing antibodies (NIEA)" is introduced, but later it's referred to as "NIEAs". 

Ensuring clarity and specificity in descriptions is crucial for the reader's understanding and the scientific accuracy of the paper. For example, in line 110, "SARS CoV-2 NTD" should be "Spike NTD"; in line 115, "NTD of SARS CoV-2" should be "NTD of SARS CoV-2 spike protein". It is essential for the authors to review the entire manuscript to ensure clarity and precision in its content.

Author Response

Response to Reviewer 2 Comments

  1. Summary

Thank you for giving us the opportunity to submit a revised version of the manuscript “Enhancement of SARS-CoV-2 Infection via Crosslinking of Adjacent Spike Proteins by NTD-Targeting Antibodies” for publication in Viruses. We thank you for taking the time to review this manuscript. We have incorporated suggestions made by reviewer and highlighted those changes within the revised manuscript. Below, please find the detailed responses below and the corresponding revisions highlighted in revised manuscript. All page and line numbers refer to the revised manuscript.

Comments 1: What impact does this research have on the development of vaccines and treatment of COVID-19? The paper would benefit from a more in-depth discussion of the implications of its findings for vaccine and therapeutic antibody development.

Response 1: Thank you for the suggestion to discuss the impact of this study for the development of vaccines and treatment of COVID-19. We have added sentences to Introduction and Discussion as follows:

Introduction (Line 73-77, 80-82):

Because ADE can have profound consequences on the safety profile of vaccines, as seen in Sanofi Pasteur’s Dengvaxia which sensitizes dengue seronegative resulting in higher risk of severe dengue

Taken together, ADE mechanism in SARS CoV-2 needs to be studied and evaluated further to be taken into considerations for possible ADE risk in developing SARS CoV-2 vaccines and therapies

Discussion (Line 335-338):

As previous study points out that antibodies generated from vaccination or from therapeutic antibody both have the potential to cause ADE of SARS CoV-2 infection in vitro [18], it is important to consider checking for ADE potential in COVID-19 therapeutic and vaccine development.

Comments 2: The study points out that the composition of NTD loop 1 in SARS-CoV-2 is distinct from that of SARS-CoV-1. Did the authors examine other coronaviruses to determine if this characteristic is exclusive to SARS-CoV-2?

Response 2: Thank you for your question. In previous study it has been described that loop 1 of SARS CoV-2 is distinct from SARS CoV and other coronaviruses from genus Betacoronavirus subgenus Sarbecovirus, yet to the best of our knowledge there are no studies comparing SARS CoV-2 NTD loop 1 with coronaviruses from other genus. We added sentences for clarity.

Result (Line 290-292):

In this regard, it is noteworthy that the composition of loop 1 in SARS-CoV-2 is completely distinct from that of SARS-CoV and other coronaviruses in subgenus Sarbecovirus [47].

Comments 3: The study suggests that NIEAs may induce allosteric changes in the NTDs, affecting the RBD transition from the “down” to the “up” state. This remains a hypothesis. Could the author offer additional evidence to support the role of these interactions?

Response 3: Thank you for the suggestions. We added sentences to support the hypothesis. NTD-RBD decoupling will be covered as subject for separate study.

Discussion (Line 339-340):

Several studies highlight the interconnectedness and coupling of NTD and RBD [43, 48, 49]. Moreover, NTD movement is involved in RBD conformational change from “down” to “up” position

  1. Response to Comments on the Quality of English Language

Comment 1: For example, in line 97, "These images clearly show the crosslinking if NTDs in a manner that is consistent with our structural model."

The word "if" seems to be a grammar error. The authors should ensure consistent use of terms throughout the manuscript. For example, in line 79, "NTD-targeting infection enhancing antibodies (NIEA)" is introduced, but later it's referred to as "NIEAs".

Response 1: We thank you for the constructive feedback. We have revised the sentence as follow:

Introduction (Line 87-88):

We then provide direct observation of NIEA-crosslinked NTDs using negative-stain electron microscopy

Comment 2: Ensuring clarity and specificity in descriptions is crucial for the reader's understanding and the scientific accuracy of the paper.

For example, in line 110, "SARS CoV-2 NTD" should be "Spike NTD"; in line 115, "NTD of SARS CoV-2" should be "NTD of SARS CoV-2 spike protein".

It is essential for the authors to review the entire manuscript to ensure clarity and precision in its content.

Response 2: We agree that specificity and clarity is necessary for readability and scientific accuracy of the paper. We have reviewed the manuscript and made the necessary changes.

Material and Methods (Line 102-103):

The SARS CoV-2 Spike NTD (amino acids 14-333) was cloned into a pcDNA3.4 expression plasmid containing a SLAM signal sequence and His-tag at C-terminus.

Material and Methods (Line 106-107):

The pcDNA3.4 plasmid containing His-tagged NTD of SARS CoV-2 spike protein was transfected to Expi293T cells using PEI max (Polysciences).